# Stress increases hepatic release of lipocalin 2 which contributes to anxiety-like behavior in mice

Lan Yan[1,8], Fengzhen Yang[1,8], Yajie Wang[1,8], Lingling Shi [1,8], Mei Wang[1], Diran Yang[1], Wenjing Wang[1], Yanbin Jia[2,3], Kwok-Fai So [1,3,4,5,6,7] & Li Zhang [1,3,5,6,7]

Chronic stress induces anxiety disorders via both neural pathways and circulating factors. Although many studies have elucidated the neural circuits involved in stress-coping behaviors, the origin and regulatory mechanism of peripheral cytokines in behavioural regulation under stress conditions are not fully understood. Here, we identified a serum cytokine, lipocalin 2 (LCN2), that was upregulated in participants with anxiety disorders. Using a mouse model of chronic restraint stress (CRS), circulating LCN2 was found to be related to stress-induced anxiety-like behaviour via modulation of neural activity in the medial prefrontal cortex (mPFC). These results suggest that stress increases hepatic LCN2 via a neural pathway, leading to disrupted cortical functions and behaviour.

Anxiety disorders affect more than 2% of the whole population[1], with a prevalence of subthreshold diseases of over 10% prevalence[2]. Multiple neuropathological mechanisms have been proposed, including monoamine neurotransmitters[3], neural circuits[4], and neuroglial cells[5]. In addition to these central factors, circulating hormones such as glucocorticoids profoundly affect mental functions[6], suggesting crosstalk between peripheral tissues and central nuclei. Recent studies have suggested the regulation of body immune functions by mood status. For example, mild stress potentiates the body immune response via amygdala-mediated splenic nerves[7]. On the other hand, emerging evidence has also revealed the participation of peripheral organs in anxiety pathogenesis, including intestinal tissues[8] and adipocytes[9]. These findings converge to imply the existence of putatively peripheral cytokines which are responsive to stress. However, the potential role of these factors in the central modulation of anxiety-like behaviors has not been completely resolved.

As the major body metabolic center, liver is closely related to mental health. Under hepatic dysfunction, psychiatric disorders may occur[10,11]. The molecular mechanism for this liver-brain axis has been attributed to metabolites such as ammonia[12] and proinflammatory cytokines[13]. In addition to classic factors such as interleukin-6 (IL-6)[14] and tumor necrosis factor-alpha (TNF-α)[15], another cytokine named lipocalin 2 (LCN2) is also secreted by hepatic tissues[16] and can profoundly affect brain functions[17]. Recent research has recognized the role of LCN2 in the pathogenesis of neurodegenerative diseases including Alzheimer's diseases (AD)[18], Parkinson's disease (PD)[19], and brain aging[20], as well as cerebral ischemia[21,22]. In human patients, elevated LCN2 was associated with depressive symptoms comorbid with heart failure[23], suggesting its potential role in mental disorders. Although brain-derived LCN2 was reported to mediate synaptogenesis and anxiety behaviors[24], there is insufficient evidence about the role of peripheral LCN2 in

[1]Key Laboratory of CNS Regeneration (Ministry of Education), Guangdong-Hong Kong-Macau Institute of CNS Regeneration, Jinan University, Guangzhou, China. [2]The First Affiliated Hospital, Jinan University, Guangzhou, China. [3]Institute of Clinical Research for Mental Health, Jinan University, Guangzhou, China. [4]State Key Laboratory of Brain and Cognitive Science, Li Ka Shing Faculty of Medicine, The University of Hong Kong, Hong Kong SAR, China. [5]Center for Brain Science and Brain-Inspired Intelligence, Guangdong-Hong Kong-Macao Greater Bay Area, Guangzhou, China. [6]Neuroscience and Neurorehabilitation Institute, University of Health and Rehabilitation Sciences, Qingdao, China. [7]Center for Exercise and Brain Science, School of Psychology, Shanghai University of Sport, Shanghai, China. [8]These authors contributed equally: Lan Yan, Fengzhen Yang, Yajie Wang, Lingling Shi. ✉e-mail: zhangli@jnu.edu.cn

mental functions, and its potential regulatory pathway also remains unclear.

In the current study, we first identified elevated serum LCN2 levels in anxiety disorder patients and chronic restraint stress (CRS) model mice with anxiety-like behaviors. Both peripheral and central blockade of the LCN2 pathway effectively relieved CRS-induced behavioral deficits. Further evidence attributed the anxiogenic effect of LCN2 to its modulation of neuronal activities in the medial prefrontal cortex (mPFC). To identify the peripheral source and regulatory pathway, hepatic tissues were recognized as the primary source of serum LCN2 under CRS, via the vagal efferent pathway originating from dorsal motor vagal nucleus (DMX). These results collectively established a previously unrecognized vagal-hepatic-cortical loop for the modulation of anxiety-like behaviors.

## Results

### Circulating LCN2 induces anxiety disorders in humans and rodents

We first recruited a small cohort of major depressive disorder (MDD) patients with prominent anxiety symptoms for blood sampling (Fig. 1a; 13 MDD patients and 13 healthy controls; see Table S1 for demographic profiles). Prior to antidepressant medication, serum assays showed more than 2-fold higher LCN2 levels in patients than age- and sex-matched healthy controls (Fig. 1b). Interestingly, individual LCN2 concentrations were inversely correlated with the Hamilton Anxiety Scale (HAMA, Fig. 1c), and their LCN2 levels were markedly decreased in all of 13 patients after 3-month treatment (Fig. 1d). These data support the potential role of LCN2 in the pathogenesis of anxiety disorders.

Subsequently, a mouse CRS model was employed to mimic human anxiety disorders (Fig. 1e), as validated through a battery of behavioral phenotyping showing unchanged locomotor behavior (Fig. S1a–c) but avoidance of the central zone in the open field, the open arm in the elevated plus-maze or the lightbox (Fig. 1f–h). Similar to human patients, CRS model mice displayed elevated serum LCN2 levels compared to those of their naïve littermates (Fig. 1i). Moreover, higher LCN2 protein concentrations were also found in cerebrospinal fluid (CSF; Fig. 1j), suggesting a correlation between central and circulating LCN2 levels. When the mice received fluoxetine treatment, peripheral and central LCN2 were both suppressed (Fig. 1l, j) in association with the relief of anxiety-like behaviors (Fig. 1f–h). Such elevation of peripheral and central LCN2 was replicated in mice receiving chronically unpredicted mild stress (CUMS; Fig. S1d–i) and had no gender-specificity (Fig. S1j–q). Therefore, peripheral LCN2 potentially contributes to anxiety-like behaviors under stress.

To further validate the role of blood-borne LCN2 in the anxiogenic process, we treated male mice with recombinant LCN2 protein for 14 days (Fig. S2a). Behavioral tests showed that peripheral LCN2 may enter the brain (Fig. S2b) and induced anxiety-like behaviors in naïve, stress-free mice (Fig. S2c–g). On the other hand, the necessity of circulating LCN2 in the anxiogenic process was tested by daily intraperitoneal infusion of polyclonal antibody (Ab-LCN2, Fig. 1k). This blockade markedly decreased serum and central LCN2 concentrations in CRS mice (Fig. 1l, m) and attenuated anxiety-like behaviors (Fig. 1n–p; Fig. S2h–j). These results collectively demonstrated the role of circulating LCN2 in eliciting anxiety disorders under stress.

### Central LCN2 disrupts neuronal activity in the mPFC to induce anxiety-like behaviors

Having observed the critical role of peripheral LCN2 in mediating anxiety disorders, we further examined the specific brain region that is responsive to LCN2 under CRS scheme. We first demonstrated the permeability of circulating LCN2 across the blood−brain barrier (BBB) by intravenous injection of recombinant LCN2 protein conjugated to Alexa Fluor 488, resulting in fluorescent signals on mPFC cells

expressing LCN2 receptor, SLC22A17 (Fig. 2a). To better mimic the central effect of LCN2, we developed an adeno-associated virus (AAV) vector expressing LCN2 under the universal promoter (EF1α) and separately injected it into 3 regions closely related to anxiety: the hippocampus, amygdala and mPFC. Although the viral vector markedly elevated LCN2 expression in target regions, no significant behavioral effect was observed when examining the dorsal hippocampus (Fig. S3a–h) or basolateral amygdala (BLA; Fig. S3i–p). In the mPFC, however, the local over-expression of LCN2 (Fig. 2b–d) effectively induced anxiety-like behaviors in naïve, unstressed mice (Fig. 2e–g) but did not alter the general locomotor behaviors (Fig. S3q–s), highlighting the anxiogenic role of LCN2 preferentially in the mPFC. As further evidence to support the specificity of the LCN2 axis, a cannula was implanted into the mPFC and connected to a mini-osmotic pump filled with a polyclonal antibody against LCN2 (Ab-LCN2; Fig. 2h, i). A 14-day persistent infusion of Ab-LCN2 markedly decreased local protein levels in the mPFC even with viral-mediated over-expression (Fig. 2j) and effectively prevented the occurrence of anxiety-like behaviors (Fig. 2k–m; Fig. S3t–v). In sum, LCN2 mainly acts in the mPFC to direct anxiogenic effects.

In an attempt to identify the neural substrate for behavioral phenotypes under LCN2 administration, we performed in vivo 2-photon calcium recording by co-injecting genetically coded calcium indicator GCaMP6s and AAV-LCN2 into the mPFC (Fig. 3a, b). The results showed that LCN2 prominently inhibited the calcium transients of pyramidal neurons (PNs) in terms of both total calcium strengths and peak values (Fig. 3c–f). These data suggested that LCN2 disrupted mPFC neuronal activity, probably contributing to anxiety disorders.

To identify the associated molecular mechanisms, we noticed that the LCN2 receptor SLC22A17 was prominently expressed in mPFC neurons (NeuN+) under the CRS paradigm, whilst another known LCN2 receptor MC4R[25] did not show significant change (Fig. 3g–i). We next investigated the functioning site of LCN2 and its receptor by employing a short hairpin RNA (shRNA) targeting the *Slc22a17* gene. The genetic knockdown manipulation of SLC22A17 in mPFC effectively attenuated LCN2-mediated anxiety-like behaviors (Fig. S4a–h), whilst the receptor ablation in hippocampus or BLA did not prevented the LCN2 effect (Fig. S4i–x). These results converged to illustrate the anxiogenic effect of LCN2 in mPFC.

We next monitored the neuronal activity mediated by LCN2-SLC22A17 specific binding. The quantification of cFos immunofluorescence suggested the suppression of mPFC neurons by LCN2, and such effects were blocked by local receptor deficiency (Fig. S5a, b). We further combined the genetic manipulation of *Slc22a17* with GCaMP6s infection to track the in vivo neural activity of the mPFC when the mouse faced stress on the elevated plus-maze (Fig. 3j, k). In control mice, neurons presented an elevation of calcium activities upon the transition from closed arm to open arm, whilst such calcium peaks were largely repressed in CRS mice and were recovered with the knockdown of *Slc22a17* gene (Fig. 3l, m). These observations agreed with following calcium recording in head-fixed awake mice under 2-photon imaging, in which silencing the LCN2 receptor did not change cortical calcium activity in naïve mice, but did prevented hypoactivity under the CRS scheme (Fig. S5c–h). In summary, LCN2 acts on its receptor in mPFC neurons to disrupt neural activity, leading to anxiety disorders.

### Hepatic LCN2 is centrally anxiogenic under stress

Although the neural mechanism of LCN2 during stress has been illustrated by our studies, the source of circulating LCN2 remains unsolved. To address this issue, we sampled multiple organs to determine their LCN2 expressional profiles. At either the transcript (Fig. 4a) or protein level (Fig. 4b), liver tissues presented a multi-fold increase of LCN2 expression upon CRS compared to naïve conditions, whilst different brain regions did not show elevated LCN2 levels.

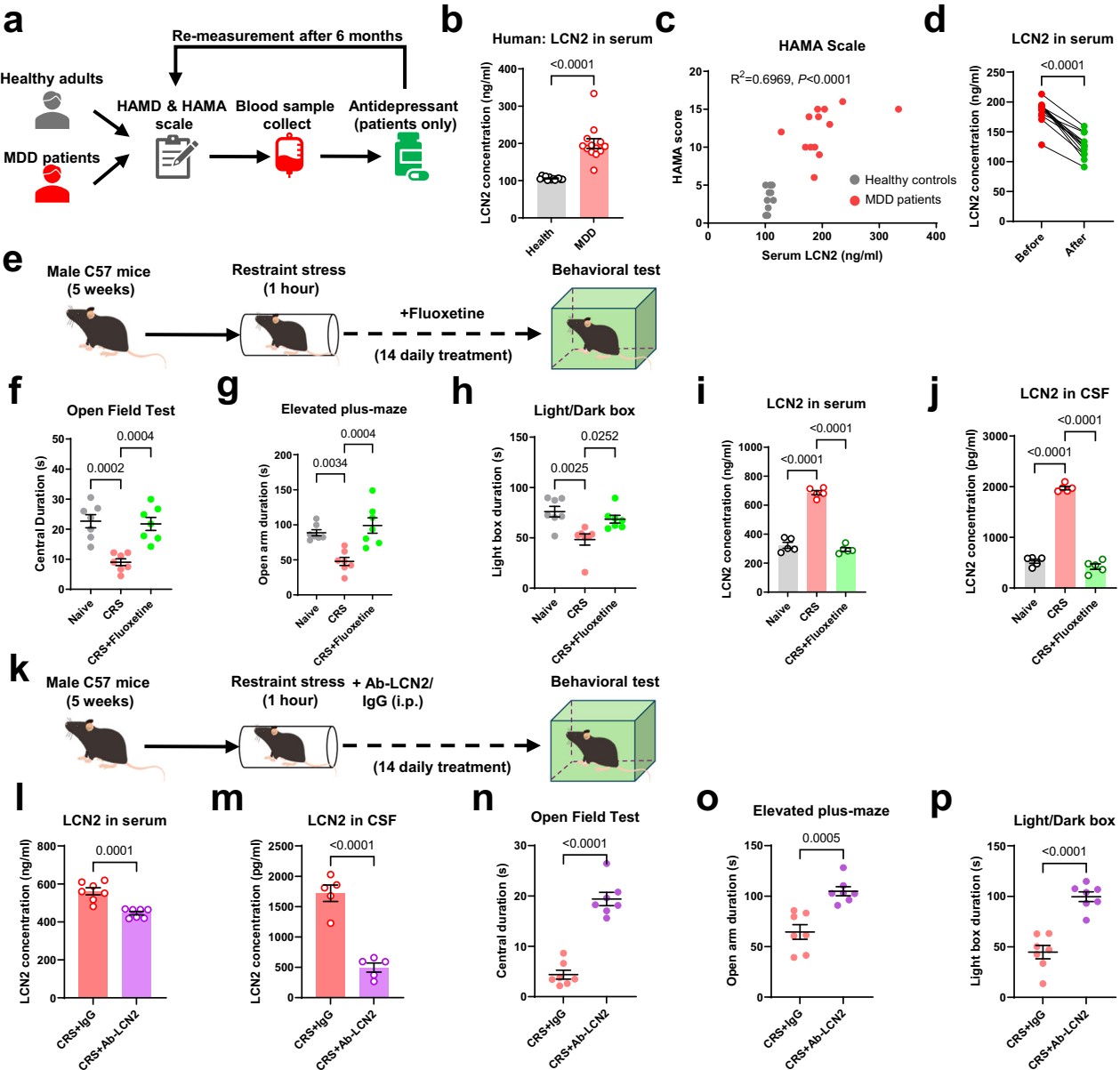

**Fig. 1 | Circulating LCN2 is related with anxiety disorders in both humans and mouse model. a** Experimental design of human cohort study, in which both healthy and major depressive disorder (MDD) patients were recruited for anxiety scaling and serum collection. **b** MDD patients exhibited higher blood LCN2 concentrations compared to healthy controls. Two-sided unpaired student $t$-test, $t(24) = 7.106$, $P < 0.0001$. **c** Serum LCN2 level is positively related with the severity of anxiety, as displayed by Hamilton Anxiety Scale (HAMA). Simple linear regression, $R^2 = 0.6969$, $P < 0.0001$. $N = 13$ individuals in each group, in (**b, c**). **d** After 3-month antidepressant treatment, 13 patients showed decreased serum LCN2 levels. Two-sided paired $t$-test, $t(12) = 11.92$, $P < 0.0001$. **e** Time schedule of chronic restraint stress (CRS) paradigm. **f** CRS treatment decreased time spent in the central arena, and fluoxetine treatment recovered normal behaviors. One-way ANOVA, $F(2,18) = 16.22$, $P < 0.0001$. **g** CRS Mice displayed avoidance toward the open arm during the elevated plus-maze task, and fluoxetine treatment attenuated such behavioral deficits. One-way ANOVA, $F(2,18) = 12.77$, $P = 0.0004$. **h** CRS decreased preference of the mouse in the light box. One-way ANOVA, $F(2,18) = 8.385$, $P = 0.0027$. $N = 7$ mice in each group, in (**f, h**). **i** CRS mice presented significantly higher serum LCN2 levels compared to naïve ones, whilst fluoxetine

treatment suppressed LCN2 concentration. One-way ANOVA, $F(2,12) = 175.1$, $P < 0.0001$. **j** LCN2 concentration in cerebrospinal fluid (CSF) followed similar patterns as those of serum samples. One-way ANOVA, $F(2,18) = 401.4$, $P < 0.0001$. $N = 5$ mice in each group, in (**i, j**). **k** Time schedules of peripheral LCN2 blockade assay, during which CRS mice received daily injection of polyclonal antibody against LCN2 for 14 days. **l** Antibody infusion decreased serum LCN2 levels of CRS mice. Two-sided, unpaired student $t$-test, $t(12) = 5.499$, $P = 0.0001$. $N = 7$ mice in each group. **m** LCN2 neutralization further inhibited CSF level of LCN. Two-sided, unpaired student $t$-test, $t(8) = 7.905$, $P < 0.0001$. $N = 5$ mice in each group.
**n** Antibody infusion increased time spent in the central arena of open field in CRS mice. Two-sided, unpaired student $t$-test, $t(12) = 9.378$, $P < 0.0001$. **o** Antagonist of LCN2 pathway also recovered normal open arm duration in CRS individuals. Two-sided, unpaired student $t$-test, $t(12) = 4.706$, $P = 0.0005$. **p** The light-avoidance behavior deficit was recovered by peripheral LCN2 blockade. Two-sided, unpaired student $t$-test, $t(12) = 5.776$, $P < 0.0001$. $N = 7$ mice in each group, in (**n–p**). Exact $P$-values were indicated using Tukey's post-hoc comparison in (**f–j**). All data were presented as mean ± sem. Source data are provided as a Source Data file.

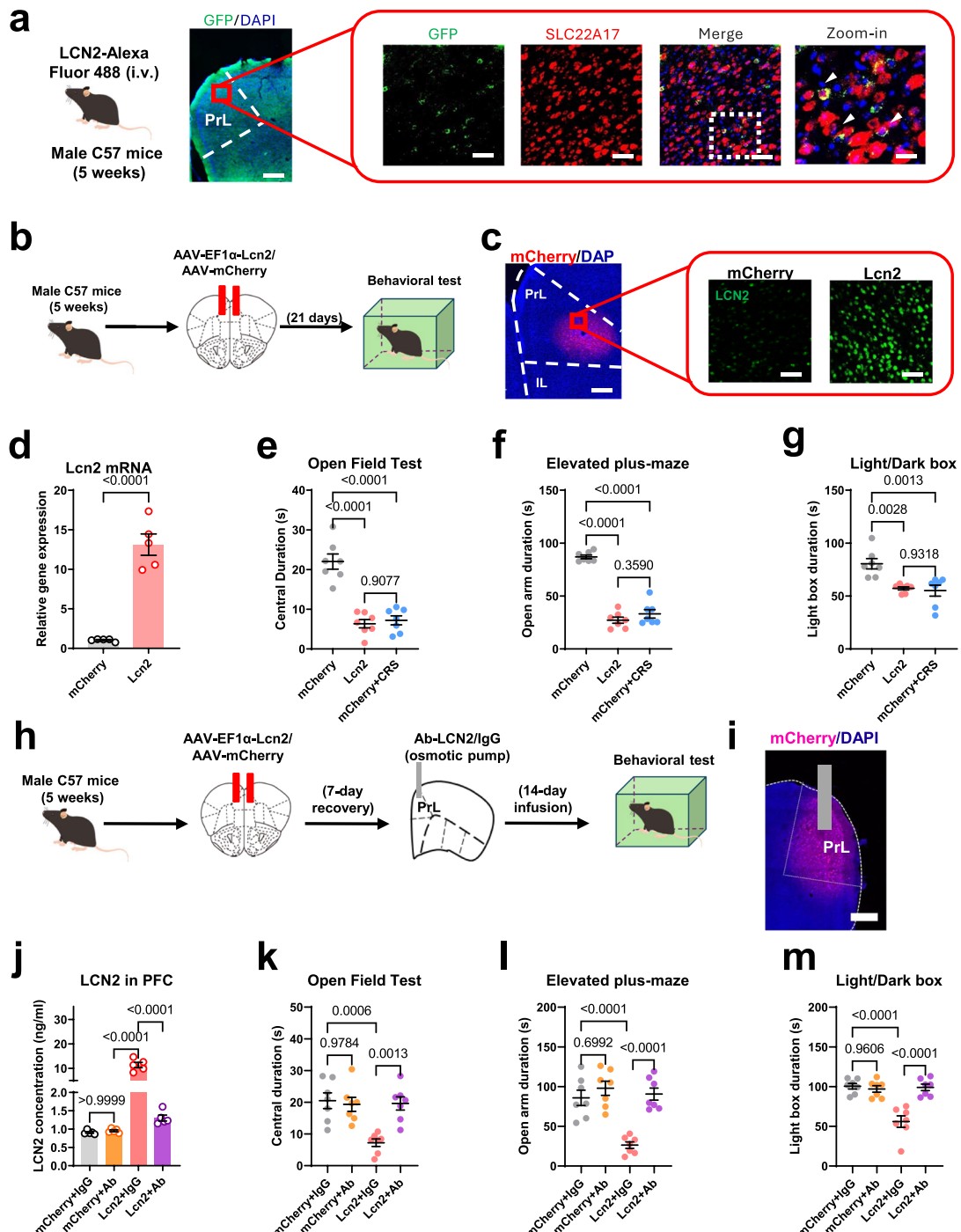

**Fig. 2 | LCN2 acts in mPFC to direct anxiety-like behaviors. a** Intravenous injection of recombinant LCN2-Alxa Fluor 488 resulted in specific protein binding onto cells expressing SLC22A17 in prelimbic (PrL) region. Triplicated studies were performed. Scale bar, 200 μm on left, and 100 μm on right. **b** Schematic diagrams of LCN2 over-expression in mPFC. Mice received an AAV vector expressing *Lcn2* gene. **c** Left panel, viral transfection site within prelimbic (PrL) region of mPFC. Right, LCN2 protein expression. Scale bar, 200 μm on left, and 100 μm on right. **d** Viral infection remarkably enhanced local gene expression of *Lcn2*. Two-sided unpaired *t*-test, $t(8) = 8.953$, $P < 0.0001$. $N = 5$ mice in each group. **e** As similar with CRS treatment, *Lcn2* gene over-expression in mPFC resulted in lower time duration in the open field. One-way ANOVA, $F(2,18) = 38.69$, $P < 0.0001$. **f** Anxiety-like phenotypes were identified as lower time spent on the open arm in both AAV-LCN2 and CRS groups. One-way ANOVA, $F(2,18) = 121.0$, $P < 0.0001$. **g** Avoidance toward the light box were observed with LCN2 transfection. One-way ANOVA, $F(2,18) = 11.19$,

$P = 0.0007$. $N = 7$ mice in each group in (**e**–**g**). **h** Experimental outlines for LCN2 blocking assay, in which Ab-LCN2 was directly applied on mouse mPFC with over-expression of *Lcn2* gene. **i** Fluorescent image showed the **i**nfection of AAV-LCN2. Triplicated studies were performed. Scale bar, 200 μm. **j** Ab-LCN2 remarkably decreased LCN2 protein contents in mPFC. One-way ANOVA, $F(3,16) = 87.50$, $P < 0.0001$. $N = 5$ mice in each group. **k** Ab-LCN2 infusion prevented the behavioral deficits induced by mPFC-specific LCN2 over-expression. One-way ANOVA, $F(3,24) = 9.571$, $P = 0.0002$. **l** The abnormal behavior in the elevated plus-maze induced by AAV-LCN2 was rescued after Ab-LCN2 administration. One-way ANOVA, $F(3,24) = 17.60$, $P < 0.0001$. **m** The light-avoidance deficit was also prevented under LCN2 neutralization. One-way ANOVA, $F(3,24) = 18.88$, $P < 0.0001$. $N = 7$ mice in each group in (**k**–**m**). Exact *P*-values were indicated using Tukey's post-hoc comparison in (**e**–**m**). All data were presented as mean ± sem. Source data are provided as a Source Data file.

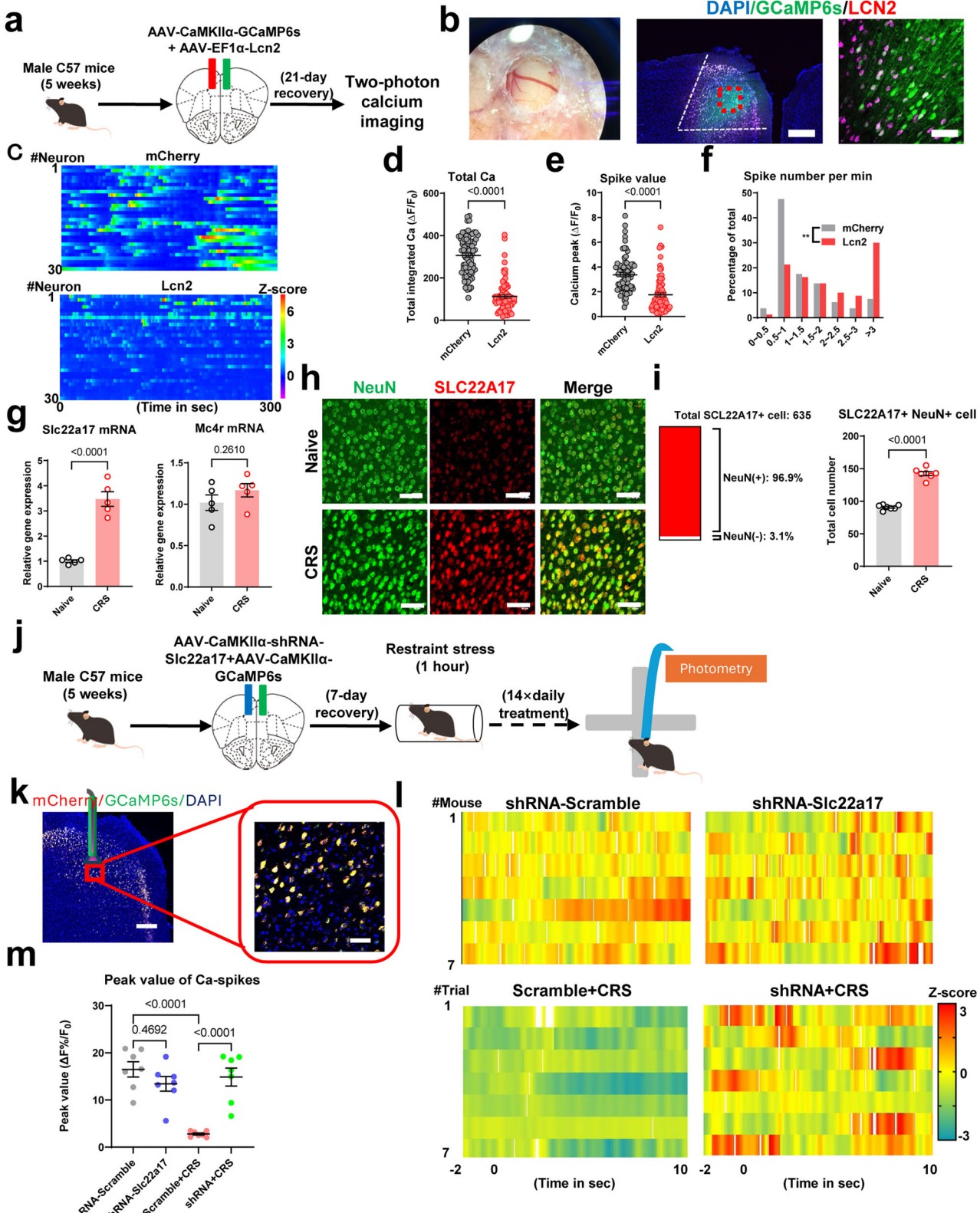

Therefore, liver might be the prominent source of stress-induced circulating LCN2. To further demonstrate the causal relationship between hepatic LCN2 and mental dysfunction, we utilized an AAV vector carrying shRNA targeting the *Lcn2* gene under the direction of the liver-specific promoter thyroxine-binding globulin (TBG). The intravenous injection of this AAV-TBG-shRNA-Lcn2 vector (Fig. 4c) successfully suppressed target gene expression in liver tissues

(Fig. 4d–f). With no effect on general health or body weight (Fig. S6a), serum (Fig. 4g) and brain (Fig. 4h) LCN2 levels were decreased with hepatic *Lcn2* gene knockdown, while the baseline level was unaffected. Behavioral assays also suggested the prevention of anxiety-like phenotypes in CRS mice with hepatic *Lcn2* gene knockdown (Fig. 4i–k). To determine the detailed mechanism, histological and molecular assays both suggested the suppression of *Slc22a17* gene expression in mPFC

**Fig. 3 | LCN2 disrupted mPFC neural activity. a** Schematics of in vivo 2-photon calcium imaging, with co-infection of AAV-LCN2 and GCaMP6s into mouse mPFC. **b** Left, imaging window under bright-field microscope. Middle and right, co-expression of LCN2 and GCaMP6s. Triplicated studies were performed. Scale bar, 250 μm on left, and 100 μm. **c** Heatmap showing calcium activities of pyramidal neurons (PNs) during the 300-s imaging period. Color-coded data were normalized and presented as z-score. A total of 30 neurons (from 4 animals) were displayed. **d** LCN2 over-expression remarkably suppressed integrated calcium activities. Nonparametric Kolmogorov–Smirnov test $D = 0.7750$, $P < 0.0001$. **e** Decreased amplitudes of single calcium spike. Nonparametric Kolmogorov–Smirnov test $D = 0.5959$, $P < 0.0001$. **f** Elevated frequency of cortical pyramidal neurons after LCN2 transfection. Nonparametric Kolmogorov–Smirnov test $D = 0.3000$, $P = 0.0015$. $n = 80$ neurons (from 4 animals) in each group in (**d–f**). **g** Quantification of *Slc22a17* and *Mc4r* gene in mPFC. Two-sided unpaired *t*-test, *Slc22a17*: $t(8) = 8.394$, $P < 0.0001$; *Mc4r*: $t(8) = 1.210$, $P = 0.2610$. $N = 5$ mice each group.

**h** Expression of SLC22A17 in mPFC neurons (NeuN+). Scale bar, 100 μm. **i** Left, most of SLC22A17 proteins were expressed on neurons. Right, the number of SLC22A17+ neurons between naïve and CRS group. Two-sided unpaired *t*-test, $t(12) = 15.22$, $P < 0.0001$. $N = 7$ mice per group. **j** Schematic illustration of in vivo fiber photometry recording on the elevated plus-maze under *Slc22a17* gene knockdown. **k** Infection of shRNA and GCaMP6s expressing viruses in mPFC. Triplicated studies were performed. Scale bar, 200 μm on left, and 100 μm on right. **l** Heatmaps showing calcium activities when the mouse entered into the open arm from the closed arm. Calcium transients were normalized and shown as z-score, and each recording window covered 2 s before and 10 s after the transition. Each line represented the averaged values of 10 trials from one mouse. **m** Attenuated calcium peak in CRS mice was reversed by knocking down *Slc22a17* gene expression in mPFC. One-way ANOVA, $F(3.24) = 17.74$, $P < 0.0001$. $N = 7$ mice in each group. Exact *P*-values were indicated using Tukey's post-hoc comparison in (**m**). All data were presented as mean ± sem. Source data are provided as a Source Data file.

(Fig. S6b, c), and cFos staining (Fig. S6d) implied the restoration of mPFC activity upon peripheral deprivation of LCN2. More importantly, in vivo calcium recording showed that liver-specific LCN2 led to the recovery of normal calcium activity under the CRS paradigm (Fig. 4l–p). Taken together, these results suggest the central role of liver-derived LCN2 in responding to stress, disrupting cortical neural activity, and leading to anxiety-like behaviors.

### The vagal efferent pathway is involved in hepatic LCN2 biosynthesis under chronic stress

We next investigated the mechanism for transducing stress into liver LCN2 biosynthesis. Previous studies have identified liver LCN2 production upon stimulation with proinflammatory factors such as interleukin-6 (IL-6)[26]. On the other hand, cytokine production in the liver is mediated by the vagus nerve[27]. We thus investigated these two possibilities (Fig. 5a) and found no significant change in multiple inflammatory cytokines in the serum of CRS mice (Fig. 5b). In addition, the signaling pathway of IL-6, including signal transducer and activator of transcription 3 (STAT3), remained intact in hepatocytes (Fig. 5c), largely rejecting the possibility of upstream modulation of LCN2 by inflammatory factors. We thus extended our research to the possibility of neural modulation by performing subdiaphragmatic vagotomy of the common hepatic branch, followed by CRS treatment (Fig. 5d, e). The surgery markedly inhibited the liver LCN2 surge under stress (Fig. 5f, g) and depressed circulating or central LCN2 levels (Fig. 5h, i). LCN2 axis suppression in the mPFC (Fig. 5j, k) was associated with the recovery of mPFC neuronal activity (Fig. 5l). In behavioral assays, hepatic denervation prevented anxiety-like behaviors in CRS mice (Fig. 5m–o).

Since the subdiaphragmatic vagal nerve includes both afferent and efferent pathways innervating liver as well as the stomach, pancreas, and intestine[28,29], we subsequently validated the specificity of vagal system innervating hepatic tissues. Targeting the α7 subunit of nicotinic acetylcholine receptor (α7nAChR) in vagus efferent pathway[30], we used both pharmaceutical and genetic approaches: (1) The α-bungarotoxin, one potent antagonist of α7nAChR was administrated before each restraint stress episode (Fig. S7a). As similar as those of vagotomy, blockade of α7nAChR inhibited hepatic *Lcn2* gene expression (Fig. S7b) as well as circulating and central protein levels (Fig. S7c, d). In behavioral phenotyping, pharmaceutical blockade of vagal efferent pathway nerve did not affect the locomotor activity but remarkably relieved CRS-induced anxiety-like behaviors (Fig. S7e–i). (2) We also infected shRNA targeting α7nAChR coding gene specifically in hepatic tissues (Fig. S7j–l), resulting in similar effects as those of pharmaceutical manipulations (Fig. S7m–t). In sum, evidence from surgical, pharmaceutical, and genetic manipulations demonstrated that the vagal efferent pathway mediates hepatic LCN2 biosynthesis under stress to induce anxiety-like phenotypes. However, due to the lack of direct evidence to demonstrate the innervation of liver

parenchyma by vagal efferent pathway, there is still a possibility that the hepatic LCN2 production is mediated by signals transmitted from adjacent organs such as gut, pancreas or hepatic portal vein, which are known to be innervated by the common hepatic branch[28,31].

### DMX responds to stress and stimulates LCN2 biosynthesis

The vagal efferent pathway originates from brain stem nuclei, and we thus tracked the brain regions responsible for liver LCN2 production under CRS. Using the pseudo-rabies virus (PRV) system to retrogradely label the upstream neurons of hepatic tissues, we found a prominent distribution of input cells in several brain stem nuclei including parasolitary nucleus (PSol), cuneate nucleus (Cu), external cuneate nucleus (ECu) and DMX (Fig. 6a). Among those regions, only DMX showed increased activity of cholinergic neurons (ChAT+) upon CRS (Fig. 6b–d). A previous study recognized the origination of the hepatic branch of the subdiaphragmatic vagus nerve from the DMX[32], it was thus proposed that the DMX mediates the hepatic response to stress. As functional evidence, we expressed the designer receptors exclusively activated by designer drugs (DREADD) receptor hM3Dq into DMX of naïve mice (Fig. S8a, b). The bilateral injection of DMX was applied due to the observed retrograde labeling pattern from liver tissues across both sides of DMX (Fig. 6a). The application of the receptor-ligand clozapine-N-oxide (CNO) activated the DMX (Fig. S8b, c), leading to markedly increased serum LCN2 levels (Fig. S8d) and anxiety-like behaviors in naïve, unstressed mice (Fig. S8e–g). More importantly, mice pretreated with vagotomy of the hepatic branch were resilient to chemogenetic activation, as shown by unchanged serum LCN2 (Fig. S8d) and the absence of anxiety-like behaviors upon DMX stimulation (Fig. S8e–g).

Since DMX-mediated vagal efferent pathways innervate multiple organs, we next manipulated hepatic-specific descending pathways by combining a hepatic infection of retrograde transsynaptic AAV-WGA-Cre[33] plus DMX-targeted expression of Cre-dependent channelrhodopsin-2 (AAV-dio-hChR2-EGFP; Fig. 6e). The liver-innervating DMX neurons seem to accumulate in the lateral part (Fig. 6f), although more systemic studies are required for further substantiations. As function evidence, the blue light stimulation in the DMX effectively elevated hepatic production of LCN2 into blood circulation (Fig. 6g, h), resulting in anxiety-like behaviors in naïve mice (Fig. 6i–k). In a similar manner, when the inhibitory light-sensitive ion channel halorhodopsin (NpHR3.0) was infected into liver-innervating DMX neurons (Fig. 6l, m), light-mediated neuronal inhibition during each CRS episode effectively relieved hepatic LCN2 surge (Fig. 6n, o) and counteracted anxiety-like behaviors under CRS but did not affect the performance of naïve mice (Fig. 6p–r). These results illustrated the specificity of a DMX-liver pathway in modulating peripheral LCN2 release and anxiety-like behaviors, although the possibility of indirect pathway relaying by gastrointestinal organs cannot be excluded.

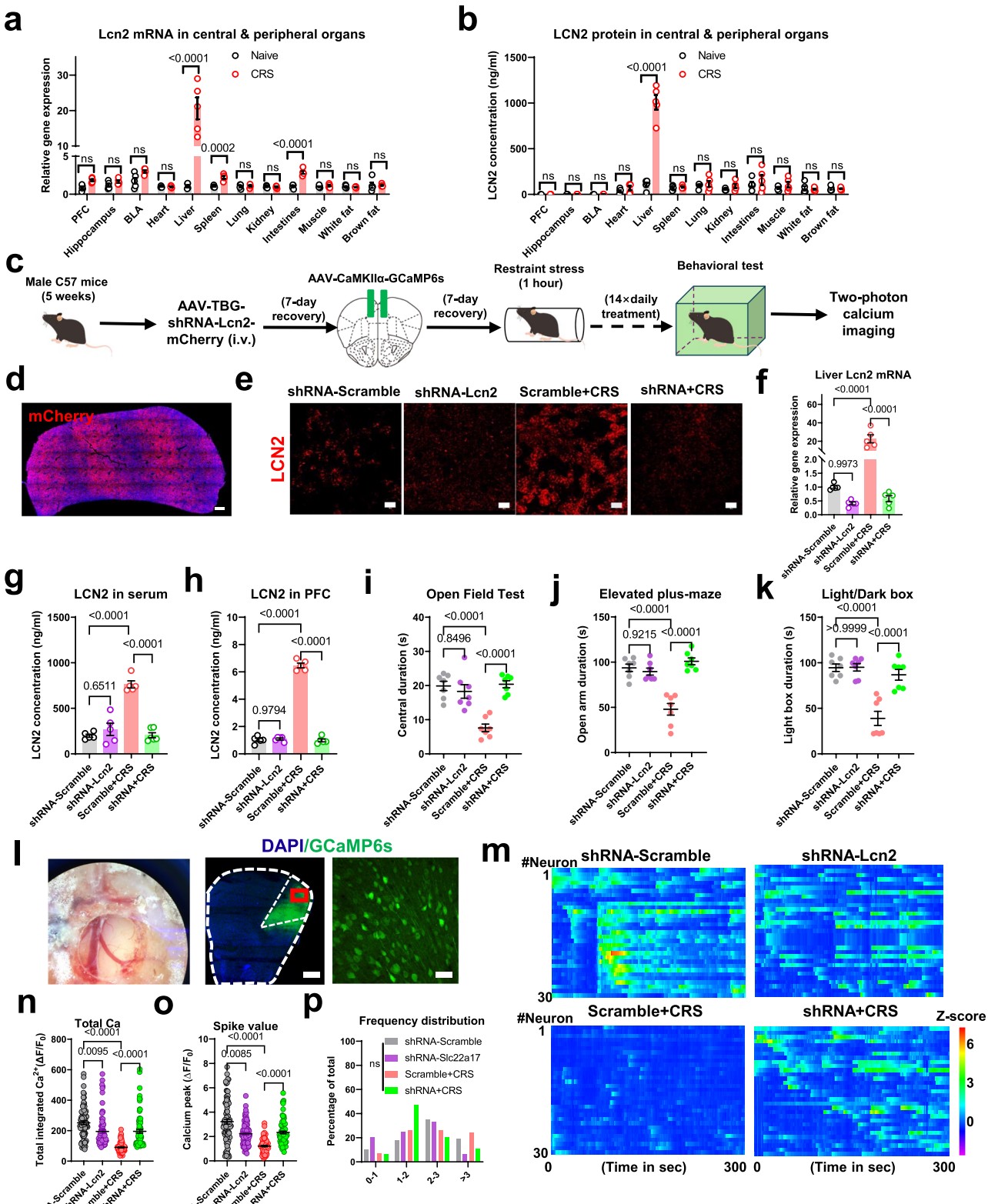

Next, we tested the specificity of LCN2 molecules within this vagal efferent modulation of anxiety-like behaviors. In DMX-activated mice, Ab-LCN2 was chronically infused (Fig. 7a). Although antibody treatment did not change the neuronal activity of the DMX (Fig. 7b, c), it did attenuate the blood LCN2 surge (Fig. 7d) and block anxiety-like behaviors following chemogenetic activation (Fig. 7e–g). Such data

proved that DMX activation induced anxiety behaviors by stimulating hepatic LCN2. On the other hand, when the inhibitory chemogenetic receptor (hM4Di) was injected into the DMX (Fig. 7h), CNO infusion largely suppressed local neuronal activity upon CRS (Fig. 7i, j) and depressed serum LCN2 levels (Fig. 7k). Behavioral assays showed consistent results, as chemogenetic inhibition improved resilience to

**Fig. 4 | Hepatic LCN2 elicits anxiety-like behaviors via neuromodulation of mPFC. a** *Lcn2* gene transcript levels were remarkably increased in liver, spleen, and intestine tissues but not in brain or other organs in CRS mice. **b** LCN2 protein was up-regulated only in liver tissues by CRS. Multiple *t*-test was used for between-group comparison in a two-sided manner, with Holm–Šídák method for correction. *N* = 5 mice in each group in (**a, b**). **c** Outlines for liver-specific LCN2 knocking down assay, in which an shRNA targeting *Lcn2* gene was introduced under hepatic promoter TBG. **d** Fluorescent image of liver tissues showing the viral infection site. Triplicated studies were performed. Scale bar, 500 μm. **e** Immunofluorescent staining of liver tissue slices for LCN2 expression. Scale bar, 50 μm. **f** shRNA transfection remarkably depressed *Lcn2* gene expression in liver tissues of CRS mice. One-way ANOVA, *F*(3,16) = 25.83, *P* < 0.0001. **g** Serum LCN2 level was also suppressed by shRNA transfection in liver. One-way ANOVA, *F*(3,16) = 42.00, *P* < 0.0001. **h** Hepatic LCN2 silencing did not affect brain LCN2 levels in naïve mice but dampened the LCN2 surge under CRS. One-way ANOVA, *F*(3,16) = 477.1, *P* < 0.0001. *N* = 5 mice in each group in (**f**–**h**). **i** Although knocking down *Lcn2* gene in liver did not affect baseline anxiety level, it did help to antagonize CRS as to increase central zone duration. One-way ANOVA, *F*(3,24) = 18.00, *P* < 0.0001. **j** Hepatic LCN2 deficiency recovered normal open arm duration. One-way ANOVA, *F*(3,24) = 25.94, *P* < 0.0001. **k** Light box avoidance behaviors in CRS animals were attenuated after hepatic LCN2 knocking down. One-way ANOVA, *F*(3,24) = 22.51, *P* < 0.0001. *N* = 7 mice in each group in (**i**–**k**. **l** Left), imaging window of 2-photon microscopy. Middle and white, infection sites of GCaMP6s. Triplicated studies were performed. Scale bar, 200 μm and 100 μm. **m** Heatmaps of relative calcium transient strengths among all groups. Data were processed as z-score and were color-coded. **n** Although the blockade of liver *Lcn2* gene expression decreased mPFC calcium level in naïve animals to some extents, it recovered normal neuronal activity in CRS group. Nonparametric Kruskal–Wallis test statistic = 140.7, *P* < 0.0001. **o** Liver *Lcn2* gene knockdown also re-elevated calcium peak values of mPFC neurons in CRS mice. Nonparametric Kruskal-Wallis test = 101.3, *P* < 0.0001. **p** Distribution of calcium transient frequency among all groups. *n* = 80 neurons (from 4 animals) in each group in (**n**–**p**). Exact *P*-values were indicated using Tukey's post-hoc comparison in (**f**–**k**), and Dunn's multiple comparison test in (**n, o**). All data were presented as mean ± sem. Source data are provided as a Source Data file.

CRS (Fig. 7l–n). The intraperitoneal injection of LCN2 did not affect DMX activity (Fig. 7i, j) but reversed the effect of chemogenetic manipulation, recapitulating anxiety-like behaviors (Fig. 7l–n).

Finally, the possibly confounding effect of hepatic LCN2 on vagal afferent pathway which ascends to the brain to induce anxiety-like behaviors[34] was tested. Different approaches were employed to verify the role of vagal afferent pathway. In the first experiment, capsaicin was locally infused onto the hepatic vagal branch (Fig. S9a) for the deafferentation of transient receptor potential vanilloid 1 (TRPV1)-neurons in vagal sensory afferents[33]. The decreased density of TRPV1+ cells in the spinal cord (Fig. S9b, c) did not rescue the anxiety-like behaviors following DMX activation (Fig. S9d–i). These results suggested that stress stimuli activate the DMX for potentiating hepatic LCN2, which induces anxiety-like behaviors by affecting mPFC functions. As an alternative approach, the nodose ganglia was labeled for the hepatic branch via injecting a retrogradely labeled AAVrg-eGFP-Cre virus into the liver, plus the Cre-dependent hM3D(Gq)-mediated chemogenetic manipulation (Fig. S10a, b). The excitation of the nodose ganglion neurons (Fig. S10c, d) did not affect liver LCN2 level or induce anxiety-like behaviors in naïve mice (Fig. S10e–j). More importantly, chemogenetic inhibition of hepatic afferent nodose ganglia during CRS episodes (Fig. S10k–n) cannot reverse the liver LCN2 surge (Fig. S10o) and did not help to relieve anxiety-like behaviors in CRS mice (Fig. S10p–t). Moreover, other vagal afferent projecting brain regions including NTS and AP were also selectively activated by chemogenetic approaches, which did not elicit circulating LCN2 peak or anxiety-like behaviors (Fig. S11). Taken together, these results show that stress activates the DMX, which is involved in hepatic LCN2 biosynthesis probably via vagal efferent pathways, further disrupting mPFC functions and resulting in anxiety-like behaviors. Our experiments cannot fully describe this brain stem-liver-cortical loop, as current evidence did not identify the signal propagation from vagal preganglionic neuron terminals to LCN-producing hepatocytes. However, these results still postulate the potentially critical and previously underappreciated role of liver-brain crosstalk in mental disorders under stress.

## Discussion

To our knowledge, this is the first systemic study revealing the role of peripheral LCN2 in anxiety disorders, although brain-derived LCN2 has been recognized to affect mental status[24,35]. Our observations demonstrated the role of LCN2 as a peripheral factor in the response to stress. Further dissections of functional pathways revealed the existence of a putative brain-liver axis whose activity is dependent on the vagal efferent nerve to mediate liver biosynthesis of LCN2. Such evidence formed a putative vagal-liver-cortex loop in which the brain stem nuclei transduced psychological stress into liver production of LCN2, which stimulated the mPFC to affect neural activity, resulting in anxiety-like behaviors.

Psychological stress induces the production of proinflammatory cytokines such as IL-6[14,36], TNF-α[37] and IL-1β[38]. These factors can be produced from neuroglial cells and act within the brain to activate neuroinflammation, leading to anxiety- or depressive-like behaviors[39–41]. On the other hand, a growing body of evidence suggests the central modulatory role of peripherally originated proinflammatory cytokines in anxiety disorders. For example, gut microbes mediate peripheral cytokine production during stress-induced anxiety[42], and traumatic stress or social defeat stress increases peripheral inflammation and reactive oxygen levels[43,44]. In peripheral organs, LCN2 plays crucial roles, as it is related to hepatic[16] or kidney[45] malfunction under acute and chronic injury and regulates metabolic inflammation in intestinal tissues[46]. Recent findings support the elevation of liver LCN2 following the production of inflammatory factors such as IL-6[26]. Our current work, however, revealed an LCN2 pathway that seems to be unrelated to those classic inflammatory factors and is dependent on vagal efferent pathway. These results support LCN2 as a potentially circulating factor for the central modulation of anxiety-like behaviors.

After revealing the regulation of mental status by circulating LCN2, we further investigated the modulatory pathway from central nuclei to peripheral organs for cytokine biosynthesis. Stress activates neural immune cells to produce proinflammatory cytokines to affect synaptic transmission[40,41] and increases sympathetic tone for the release of cytokines[47]. A recent neuroanatomical study established a brain-spleen connection that enhance humoral immunity under mild stress[7]. In the current study, we identified the potential connection between hepatic tissues and DMX, whose activation leads to elevated LCN2 expression. These findings paralleled those from established models in which modulation of the vagus nerve affects body inflammatory response and cytokine production[27]. In particular, DMX connect with the liver tissue regulates the production of TNF-α[48], and vagal nerve stimulation increases the phagocytosis of hepatic macrophages[49]. Our findings of LCN2 thus shed more insights into the central-peripheral regulatory pathway under psychological stress.

LCN2 is known as a pleiotropic factor that can be secreted by neurons and glial cells to affect brain functions[35]. The mechanistic explanations of LCN2 inside the brain include the disruption of BBB integrity[17], the activation of microglia[22,50], macrophages[51], or astrocytes[21], and axonal demyelination[52]. Although LCN2 is well known to affect neuroimmune function, the neuronal response to LCN2 has also been reported. For example, LCN2 interferes with the normal CC chemokine receptor 5 (CCR5) pathway, leading to neuronal death under human immunodeficiency virus (HIV) infection[53]. The role of LCN2 in modulating specific behaviors has been previously recognized

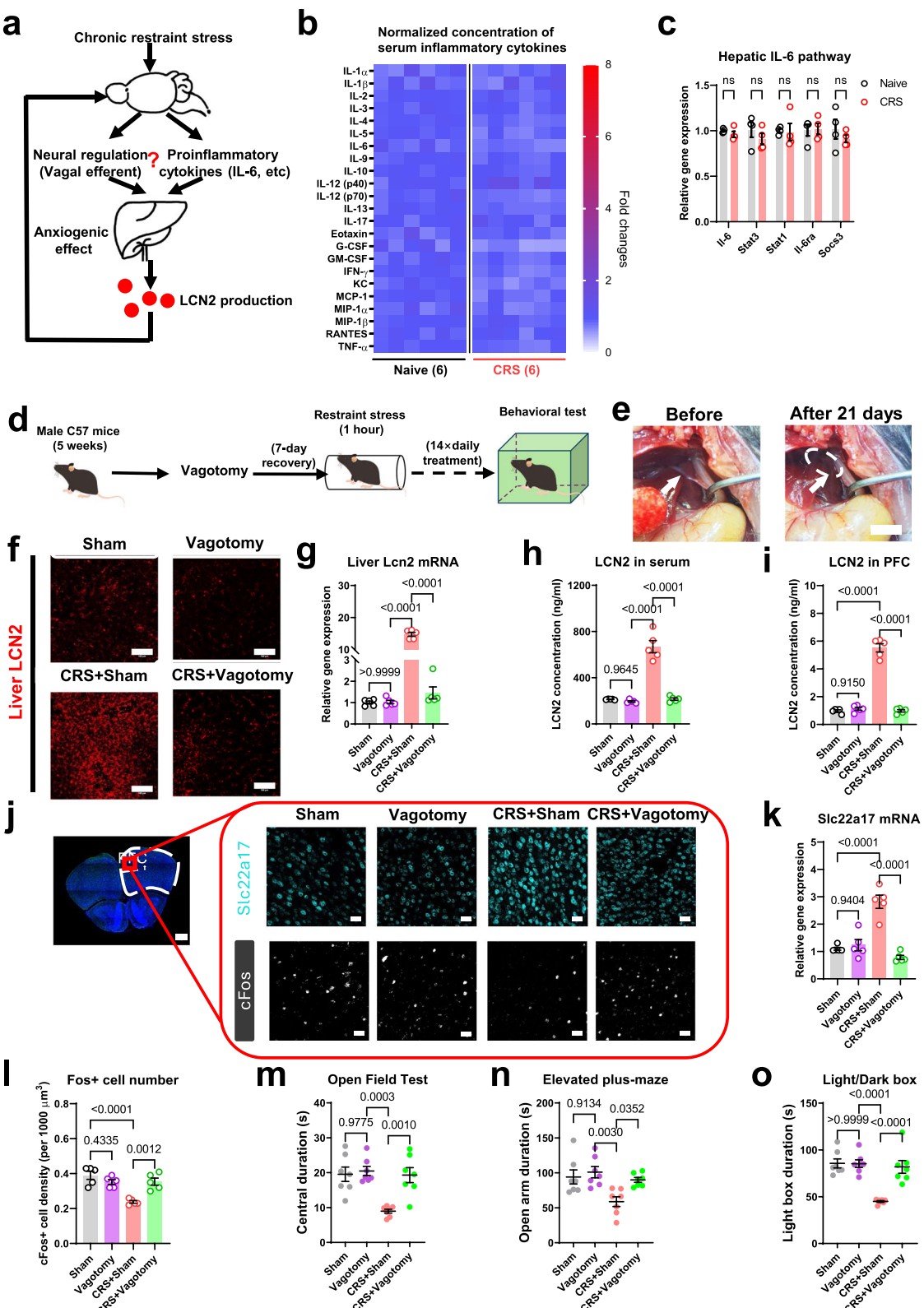

by its affinity to activate melanocortin 4 receptor (MC4R) in the hypothalamus to suppress food intake[25]. The neuromodulator function of LCN2 has its molecular substrate due to the wide distribution of SLC22A17 on neurons[50], agreeing with our observations. Our study mainly focused on the neuronal LCN2 receptor, which was shown to suppress cortical neural activity. Such a modulatory effect paralleled previous ex vivo findings showing the retraction of hippocampal

spines by LCN2[24]. These results showed that LCN2 disrupted synaptic homeostasis in terms of both structural and functional plasticity, probably contributing to anxiety disorders.

LCN2 has been reported to be related to various neurological diseases. For example, in AD patients, LCN2 levels in postmortem brain tissues were found to be elevated[18]. The potency of serum LCN2 as a biomarker for neurodegenerative diseases, however, is still

**Fig. 5 | Vagal efferent pathway evokes hepatic LCN2 to induce anxiety disorders. a** Theoretical model of pathways regulating liver LCN2 biosynthesis. **b** Relative concentrations of major serum inflammatory cytokines were unchanged between naïve and CRS mice. $N = 6$ mice in each group. **c** Undisturbed signaling pathway of IL-6 in hepatic tissues under CRS. $N = 4$ mice in each group. Stat1, signal transducer and activator of transcription 1; Il-6ra, interleukin-6 receptor alpha; Socs3, suppressor of cytokine signaling 3. Multiple $t$-test (in two-sided manner) was used for between-group comparison, with Holm–Šídák method for correction. **d** Timeline for vagotomy assay. **e** The surgical site for subdiaphragmatic vagotomy, showing the separated and truncated vagal nerves before and at 21 days after dissection. **f** Immunofluorescent staining for liver LCN2 expression after vagotomy. Scale bar, 100 μm. **g** Liver *Lcn2* gene transcript levels were suppressed by vagotomy under CRS but not in naïve condition. One-way ANOVA, $F_{(3,16)} = 400.4$, $P < 0.0001$. **h** Vagotomy did not change basal level of LCN2 but helped to relive over-expression of LCN2 in serum under CRS. One-way ANOVA, $F_{(3,16)} = 73.68$, $P < 0.0001$. **i** LCN2

levels in mPFC were also attenuated by vagotomy under CRS. One-way ANOVA, $F_{(3,16)} = 179.1$, $P < 0.0001$. **j** Immunofluorescent staining for SLC22A17 and cFos in PrL after vagotomy and CRS. Scale bars, 500 μm on left image, and 100 μm in right panels. **k** Vagotomy abolished the elevated expression of *Slc22a17* gene in CRS but not in naïve animals. One-way ANOVA, $F_{(3,16)} = 30.09$, $P < 0.0001$. **l** CRS mice showed recovery of cortical neuron activity by vagotomy. One-way ANOVA, $F_{(3,16)} = 14.48$, $P < 0.0001$. $N = 5$ mice in each group in (**g**–**l**). **m** Vagotomy did not affect baseline level of central zone duration, but antagonized anxiety-like behaviors in CRS animals. One-way ANOVA, $F_{(3,24)} = 12.21$, $P < 0.0001$. **n** Avoidance toward the open arm in CRS group was attenuated by vagotomy. One-way ANOVA, $F_{(3,24)} = 13.92$, $P < 0.0001$. **o** The duration in the light box of CRS mice was recovered after vagotomy. One-way ANOVA, $F_{(3,24)} = 24.18$, $P < 0.0001$. $N = 7$ animals in each group in (**m**–**o**). Exact $P$-values were indicated using Tukey's post-hoc comparison in (**g**–**o**). All data were presented as mean ± sem. Source data are provided as a Source Data file.

inconclusive[54]. Our study did provide plausible evidence of LCN2 as a circulating biomarker for the diagnosis of anxiety disorders, although large cohort studies are needed for further substantiation. Recent research on LCN2 in clinical trials has mainly focused on patients with cardiovascular diseases comorbid with depressive disorders[23,55]. A few other studies also observed the potency of LCN2 as an inflammatory marker for depression in elderly people[56,57]. Our study, on the other hand, recruited younger MDD patients and suggested the potential value of LCN2 in disease diagnosis. More importantly, since the improvement of anxiety symptoms was associated with decreased serum LCN2, its value as a clinical biomarker in evaluating anti-depressant efficiency can be further appreciated.

Current study has certain limitations and weakness based on available evidence. First, the direct innervation of vagal efferent pathway toward the hepatic tissues for stimulating LCN2 secretion cannot be fully demonstrated. As other gastrointestinal organs such as the duodenum, pancreas, and hepatic portal vein are known to be mediated by the common hepatic branch[28,31], we thus cannot exclude the possibility that cross-organ talk between liver and gut leads to LCN2 secretion. Nevertheless, we have employed liver-specific down-regulation of α7nAChR (Fig. S7j–t), to show the specificity of liver-targeted mediation by stress-induced DMX activation. Second, the possible role of other brain stem nuclei, such as NTS, cannot be excluded from the current model. As NTS nuclei also presented higher cFos activity upon CRS (Fig. 6b, c), and NTS has been reported to receive afferent inputs from liver tissues[28], it is possible that hepatic afferent pathway also modulated the neural network involving NTS and DMX. In future, more sophisticated studies employing cell-specific labeling, tracing, and manipulating approaches can be used to fully describe this proposed DMX-liver-prefrontal loop.

In sum, our study has demonstrated a previously unrecognized vagal-hepatic pathway whose activation under chronic stress induced liver biosynthesis of LCN2. The elevated serum LCN2 further disrupted normal neural activity in the mPFC, leading to anxiety-like behaviors. These data shed more insights into the liver-brain crosstalk under psychological stress and provide targets for the early diagnosis of or intervention for anxiety disorders.

## Methods
### Experimental animals
Male and female C57BL/6 J mice (5–6 weeks old) were purchased from Guangdong Medical Laboratory Animal Center. The exact number for each experiment was specified in figure legends. All animals were group-housed in a ventilated animal facility under normal light-dark cycle (light from 0700 to 1900) with food and water *ad libitum*. All animal experimental protocols have been pre-approved by the Ethics Committee of Experimental Animals of Jinan University in accordance with Institutional Animal Care and Use Committee guidelines for animal research.

In this study, the sample size ($N$) was selected mainly according to the exercise in similar studies. In general, $N = 7$-9 for behavioral assays, and $N = 4$-6 for molecular studies. Exact number of animals in each experiment has been reported in the relevant figure legends. No exclusion was performed for all studies, except those required by the animal ethical code. Due to the absence of gender-specific effect of LCN2 (Fig. S1j–q), male mice were adopted in most of assays.

### Collection of human blood sample
Ethical approval for this study was obtained from the Research Ethics Board of Research Ethics Committee of the First Affiliated Hospital of Jinan University (Guangzhou, China). Informed consents have been obtained in written form from all participants before conducting the assay. The blood samples were obtained from patients with major depressive disorders (5 males and 8 females) and healthy individuals (6 males and 7 females), with no known exposure to medicines within the last week. Clinical diagnosis of depression was established according to a Structured Clinical Interview for DSM-5 criteria (2013). Control subjects were age-matched, with normal cognitive function and no prior history of mental disorder or vital organ dysfunction. Due to the relatively small sample size, the sex-dependent analysis was not considered here.

Plasma was prepared from venous blood collected in EDTA-containing tubes and was then centrifuged at $1000 \times g$. The serum was collected and stored at −80 °C until use. Before anticipation, EDTA was removed using a 3.5 kDa D catheter dialyzer plasma (Millipore, US) in PBS.

### Mouse stress model and behavioral tests
In CRS model, mice were restrained in a conical plastic bag with ventilation holes for 1 h (from 7 to 8 pm) per day in 14 consecutive days. In CUMS model, mice received following treatment in a randomized order for 14 days, and different stimuli were applied in any of 2 consecutive days. Mild stress include: (1) 5 min foot electric shock; (2) Food deprivation for 24 h; (3) Cage tilt for 8 h; (4) 12-h bright light treatment in the night; (5) 3-h restraint stress; (6) 30 min stimuli from the predator odors; (7) Singly housed for 24 h; (8) 5 min of forced swimming in the cold water.

**Open field test.** The open field test was carried out in a rectangular chamber ($40 \times 40 \times 30$ cm) composed of gray polyvinyl chloride with a 25 W halogen bulb (180 cm above the field). The test mouse was gently placed on the center and was allowed to freely explore for 5 min. Images of the paths were stored and locomotor activity was analyzed using EthoVision 7.0 software.

**Elevated plus-maze test.** Mice were acclimatized in the test room for 1 h before testing. At the beginning, the mouse was gently placed in the central platform ($5 \times 5$ cm) of the elevated plus-maze and was allowed free to explore the two opposing open arms ($30 \times 5 \times 0.5$ cm) and two

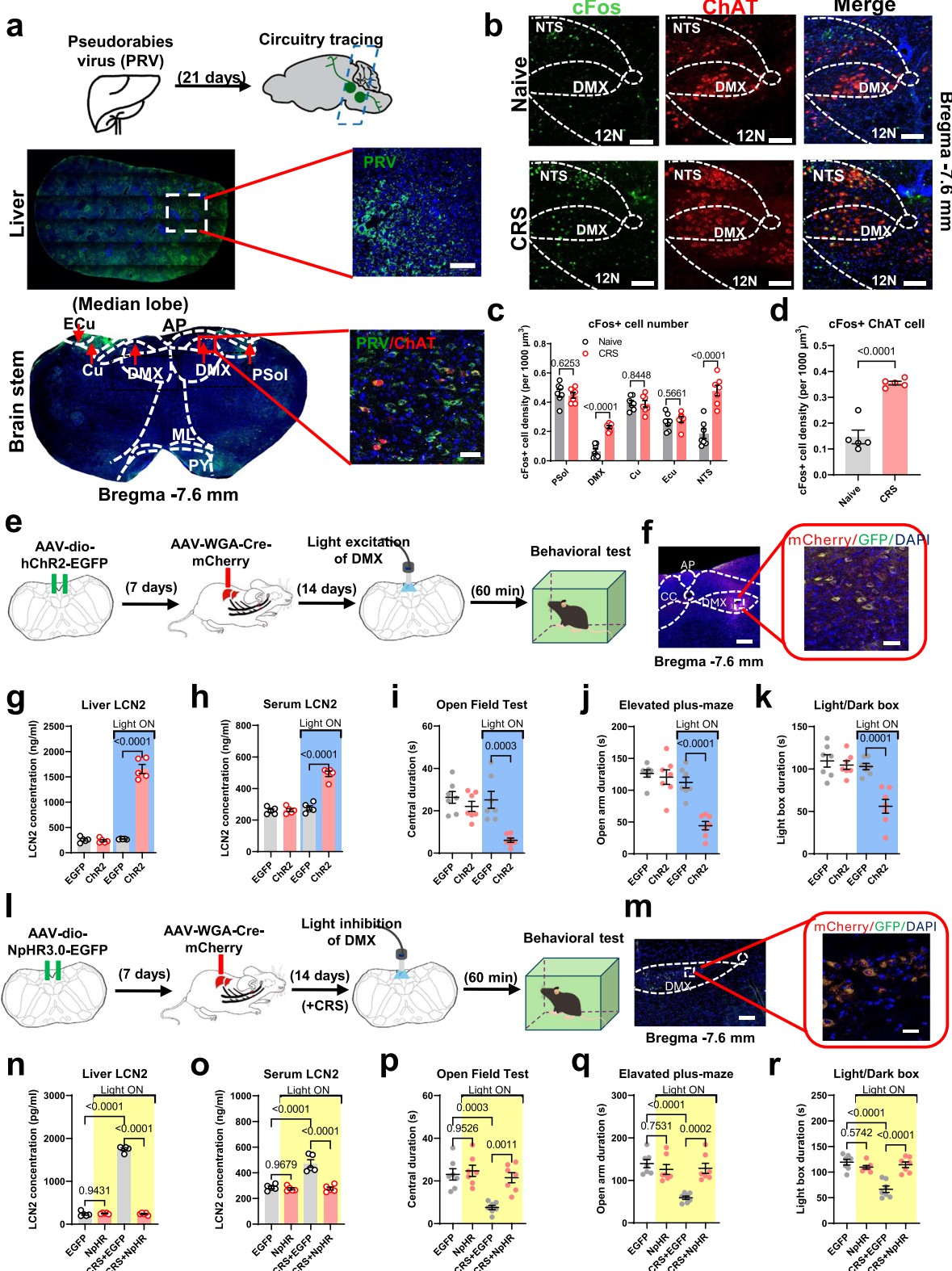

opposing enclosed arms (30 × 5 × 15 cm) for 6 min. The duration time in the open and closed arms was tracked and recorded by a video tracking system (EthoVision, Noldus, Netherland).

**Light/dark box test.** Two boxes are (16 × 16 × 25 cm) were connected by a small hole, through which the mouse can freely shuttle between two chambers. The dark box was painted in black and covered with a black lid, and the light box was painted in white and illuminated by a 60 W light bulb. At the beginning of the experiment, the mouse was put in the middle of the white box. The duration time in the light and dark box during single 5-min recording period was analyzed by Etho-Vision package.

**Fig. 6 | DMX responses to stress and stimulates hepatic LCN2. a** Upper, Retrotracing of liver-innerved nuclei by pseudo-rabies virus (PRV) system. Middle, viral injection sites in liver tissues. Lowe panels, transsynaptic expression of PRV particles in brain stem nuclei. Triplicated studies were performed. Scale bars, 500 μm on middle pane and 100 μm on lower panel. PSol parasolitary nucleus, Cu cuneate nucleus, ECu external cuneate nucleus, AP area postrema, ML medial lemniscus, PY pyramidal tract. **b** Neuronal activation profile of DMX under CRS. 12 N, hypoglossal nucleus. Scale bar, 100 μm. **c** Quantification of cFos activity showed more neuron activations in DMX not other nuclei after CRS. Multiple *t*-test (in two-sided manner) was used for between-group comparison, with Holm–Šídák method for correction. *N* = 7 mice in each group. **d** cFos activity of ChAT+ neurons in DMX of CRS mice. Two-sided unpaired *t*-test, *t*(8) = 7.598, *P* < 0.0001. *N* = 5 mice in each group. **e** Timelines for optogenetic activation of liver-innervating DMX neurons. **f** Viral infection sites within DMX. Triplicated studies were performed. Scale bar, 250 μm (left) and 25 μm (right). **g** Light stimulation induced hepatic LCN2 biosynthesis. One-way ANOVA, *F*(3,16) = 219.2, *P* < 0.0001. **h** Circulating LCN2 was also elevated upon DMX activation. One-way ANOVA, *F*(3,16) = 78.73, *P* < 0.0001. *N* = 5 mice per group in (**g**, **h**). **i** DMX activation induced the avoidance toward the central region of

the open field in unstressed mice. One-way ANOVA, *F*(3,24) = 11.75, *P* < 0.0001. **j** DMX activation decreased time spent in the open arm of the elevated plus-maze. One-way ANOVA, *F*(3,24) = 20.93, *P* < 0.0001. **k** DMX activation led to lower preference toward the light box. One-way ANOVA, *F*(3,24) = 15.82, *P* < 0.0001. *N* = 7 mice per group in (**i–k**). **l** Timelines for optogenetic inhibition of liver-innervating DMX neurons. **m** Viral infection site within DMX. Triplicated studies were performed. Scale bar, 250 μm (left) and 25 μm (right). **n** Light-mediated neuronal inhibition prevented hepatic LCN2 surge in CRS mice. One-way ANOVA, *F*(3,16) = 1357, *P* < 0.0001. **o** Circulating LCN2 in CRS-treated animals was also repressed upon DMX inhibition. One-way ANOVA, *F*(3,16) = 30.69, *P* < 0.0001. *N* = 5 mice per group in (**n**, **o**). **p** DMX inhibition rescued the avoidance toward the central region of the open field in CRS mice. One-way ANOVA, *F*(3,24) = 12.17, *P* < 0.0001. **q** DMX inhibition recovered normal time spent in the open arm of the elevated plus-maze. One-way ANOVA, *F*(3,24) = 13.92, *P* < 0.0001. **r** DMX inhibition recovered normal preference toward the light box of CRS mice. One-way ANOVA, *F*(3,24) = 20.54, *P* < 0.0001. *N* = 7 mice per group in (**p–r**). Exact *P*-values were indicated using Tukey's post-hoc comparison in (**g–r**). All data were presented as mean ± sem. Source data are provided as a Source Data file.

### Collection of mouse serum
Mouse blood sample were collected by eyeball bleeding after deep anesthesia and were treated with EDAT. Plasma was prepared by centrifugation at $1000 \times g$, and was collected serum for −80 °C storage until use.

### Cerebrospinal fluid extraction
The mouse was deeply anesthetized, and the head skin was shaved and disinfected. The animal was fixed on an operation plate to make incision of skin and muscles on top of the foramen magnum, though which a sterilized microcapillary tube connected with a 1 ml syringe was inserted. The cerebrospinal fluid was gently sampled and collected into micro tubes on the ice. The sample was stored at -80°C refrigerator for conducting molecular assays.

### RNA extraction and qPCR
Total RNA was isolated using TRI Reagent (BBI). To quantify mRNA expression levels, equal amounts of cDNA were synthesized using the High-Capacity cDNA Reverse Transcription kit (TaKaRa). cDNA samples were then mixed with TB Green Premix Ex Taq (TaKaRa) and specific primers of target genes (*Lcn2*: Forwards, TGGCC CTGAG TGTCA TGTG; Reverse, CTCTT GTAGC TCATA GATGG TGC. *Slc22a17*: Forward, TTTGG CCGTC GTGGG ATTG; Reverse, GGCGC AATCA GGTAG ACACC. *Mc4r*: Forward, CCCGG ACGGA GGATG CTAT; Reverse, TCGCC ACGAT CACTA GAATG T. *Gapdh*: Forward, AGGTC GGTGT GAACG GATTT G; Reverse, TGTAG ACCAT GTAGT TGAGG TCA). *Gapdh* gene was amplified as an internal control. Quantitative RT-PCR was carried out in a CFX384 Real-Time System (Bio-Rad). Ct values were calculated to compare the relative expression level.

### Immunofluorescence staining
Mice were deeply anesthetized and were perfused with paraformaldehyde (PFA). The whole brain was extracted and was dehydrated with 30% sucrose overnight. Brain tissues were sectioned into 40 μm coronal slices under a sliding microtome (Leica, Germany). Tissue slices were firstly washed in PBS, followed by BSA blocking, and were incubated in dilutions of primary antibody (Table S2) at 4 °C for 48 h. Secondary antibody (listed in Table S2) was then added for incubation. Fluorescent images were captured under a confocal microscope (ZEISS, Germany).

### Stereotaxic viral injection
Mice were firstly anesthetized with 1.25% Avertin. The scalp was locally sterilized and the skin was open to remove excess periosteum tissues. The medial prefrontal cortex (PFC: AP + 2.3 mm; ML ± 0.1 mm; DV −1.2 mm; DMX: AP −7.5 mm; ML ± 0.3 mm; DV −4.2 mm; relative to the

Bregma) was located by a stereotaxic instrument (RWD, China). The skull was drilled to open an imaging window by a high-speed micro-drill (OmniDrill35, WPI, USA). A total of 100 nl (mPFC) or 50 nl (DMX) viral dilutions (>1 × 10$^{12}$ gene copies/ml, listed in Table S2) was slowly injected using a glass micropipette connected to an ultra-micro injection pump (Nanoliter 2010, World Precision Instruments, Sarasota, USA). For mPFC injection, a 60° angle was adopted to avoid the damage of the imaging site. The micropipette was retained for 10 min before retraction. The head skin was sutured for post-surgery monitor.

### In vivo 2- photon calcium imaging
Calcium imaging protocols was modified from previously documented methods[55]. At 21 days after recovery from the stereotaxic injection, mice were anesthetized with 1.25% Avertin. The scalp was incised to expose the skull, on which two customized metal bars were attached with glue (Loctite 401). At 24 h later, an imaging window (2 mm by 2 mm) was created by a high-speed micro-drill, and the cortical tissue was cover by a glass coverslip using Vetbond Tissue Adhesive (3 M, USA). The tissue imaging was performed under a water-immersed objective (20×, 1.1 numerical aperture; ZEISS, Germany) using 920 nm excitation laser under an LSM780 two-photon microscope (ZEISS, Germany). During imaging, the laser power was restricted below 25 mW. Acquired time-series images were corrected by TurboReg module of ImageJ. The fluorescent value F was quantified by average pixels extracted from designed region of interest covering identifiable soma. The $\Delta F/F_0$ was calculated as $(F - F_0)/F_0$, where the $F_0$ was averaged *F*-values during the first 10% recording period as the basal level. A calcium transient was defined when the $\Delta F/F_0$ is higher than three-fold of standard deviation above the average value.

### In vivo fiber photometry recording
Populational calcium signals in PrL were collected in a commercialized fiber photometry system (ThinkerTech Inc., China). At 7-14 days before recording, an optic fiber coated with ceramic ferrule (diameter: 2.5 mm, ThinkerTech Inc, China) was implanted into the PrL (AP: 2.33 mm; ML: 0.1 mm; DV: 0.6 mm). The base of the optical fiber was glued by a screw onto the skull. During recording, a 470 nm LED (40 mW at fiber tip) was used for excitation, while calcium-independent signals were obtained using a 410 nm LED (20 mW at fiber tip) to correct for movement artifacts. The fluorescence signals were filtered (35 Hz cut-off) and normalized to calculate the fluorescent change ($\triangle F/F_0$), where $F_0$ was the baseline fluorescent level. For in behavioral recording, each event trace was extracted with the reference to the tag (from −2 s to +10 s relative to the transition from the closed arm to the open arm). Relative fluorescence were converted to the z-score and presented as the heatmap series.

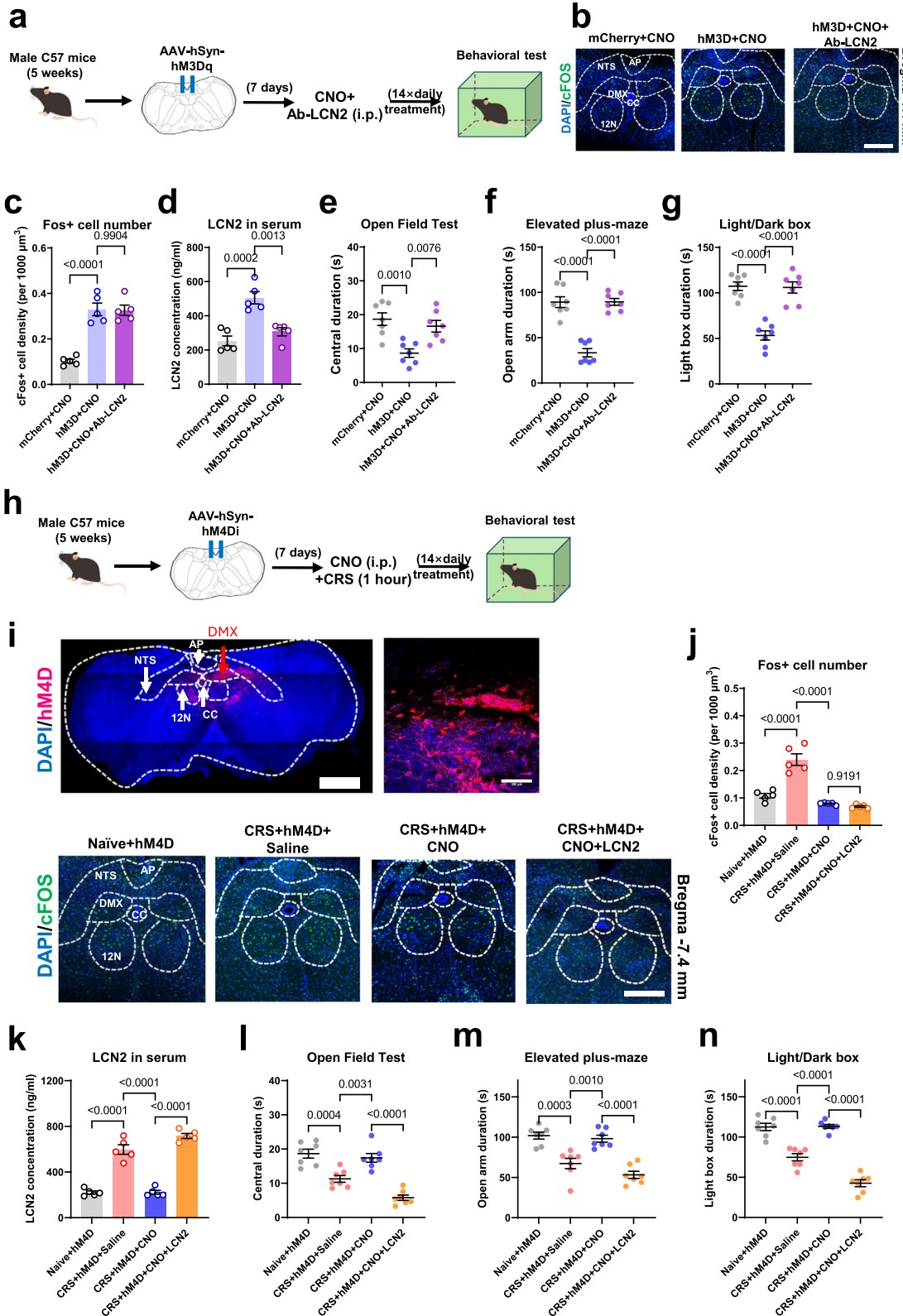

## Optogenetics stimulation

Stimulating optic fibers with ceramic tubes (diameter: 2.5 mm; length: 6 mm) were implanted into the DMX (AP −0.71 mm; ML ± 0.3 mm; DV −5.1 mm) at 7-14 days before the light stimulation, with fixation on the skull aided by micro screws. For excitation by ChR2, 10 mW light pulse (473 nm, 20 Hz, 5 ms duration) were performed for 10 min. For inhibition by NpHR3.0, a constant 10 mW light (594 nm) was adopted. The

stimulation light was provided by a power source (Changchun New Industries Optoelectronics Tech., China) under commanded by a trigger apparatus (ThinkTech Inc., China).

## Subdiaphragmatic vagotomy

Mice were anaesthetized with Avertin. The abdominal skin was disinfected, and the hair was shaved. Under a stereoscope, a midline incision

**Fig. 7 | DMX-vagal-hepatic pathway mediates anxiety-like behaviors.**
**a** Experimental designs for the peripheral blockade of LCN2 under DMX activation. **b** Immunofluorescent staining for the neuronal activity. Scale bar, 100 μm. **c** Peripheral infusion of Ab-LCN2 did not affect the DMX activation upon CNO. One-way ANOVA, $F_{(2,12)} = 37.10$, $P < 0.0001$. **d** Serum LCN2 surge by DMX activation was attenuated. One-way ANOVA, $F_{(2,12)} = 20.06$, $P = 0.0001$. $N = 5$ mice in each group in (**c**, **d**). **e** LCN2 neutralization prevented anxiety-like behaviors under DMX activation. One-way ANOVA, $F_{(2,18)} = 10.61$, $P = 0.0009$. **f** Avoidance toward the open arm after DMX-activation was rescued by Ab-LCN2. One-way ANOVA, $F_{(2,18)} = 44.92$, $P < 0.0001$. **g** Duration in the light box was re-elevated by peripheral neutralization of LCN2. One-way ANOVA, $F_{(2,18)} = 33.61$, $P < 0.0001$. $N = 7$ mice in each group in **e**–**g**. **h** Experimental design for DMX inhibition assay. **i** Upper, viral infection site. Lower, cFos staining under CNO infusion. Scale bars, 500 μm in the upper left, 100 μm in upper right and lower panels. **j** CNO administration suppressed DMX activation under CRS. One-way ANOVA, $F_{(3,16)} = 46.66$, $P < 0.0001$. **k** CRS-induced serum LCN2 peaks were attenuated by DMX inhibition. One-way ANOVA, $F_{(3,16)} = 98.20$, $P < 0.0001$. $N = 5$ mice in each group in (**j**, **k**). **l** Decreased time spent in the central zone under CRS was reversed by DMX inhibition, which can be abolished following peripheral LCN2 administration. One-way ANOVA, $F_{(3,24)} = 29.53$, $P < 0.0001$. **m** CRS-induced avoidance toward the open arm was attenuated under DMX inhibition and was reoccurred by LCN2 infusion. One-way ANOVA, One-way ANOVA, $F_{(3,24)} = 23.07$, $P < 0.0001$. **n** Time spent in the light box was recovered by DMX inhibition and was impaired under LCN2 injection. One-way ANOVA, $F_{(3,24)} = 71.44$, $P < 0.0001$. $N = 7$ mice in each group in (**l**–**n**). Exact $P$-values were indicated using Tukey's post-hoc comparison. All data were presented as mean ± sem. Source data are provided as a Source Data file.

(0.5 ×2 cm) was made to separate the skin and muscle for exposing the abdominal cavity. The gastrohepatic ligament was identified and separated using fine forceps. The connective tissue between the stomach, esophagus, and liver was gently separated to locate the ventral subdiaphragmatic vagal trunk. The hepatic branch of this vagal nerve was cut with aseptic surgical scissors. The abdominal cavity was closed with skin sutured. The mouse was returned to the home cage for 7-day recovery. For the sham control group, mice received the same surgery, in which the vagal nerve was gently exposed but without cut.

## Capsaicin-mediated vagal afferent nerve inhibition
Hepatic afferent vagus nerve was separated and exposed as described above. A cotton rod soaked with vehicle (Tween 80: olive oil = 1:9 in v/v) or 10 mg/ml capsaicin was directly applied onto the vagal nerve for 30 min. The abdominal incision was closed, and the mouse was returned to the home cage for 7-day recovery.

## Statistical analysis
Data samples were tested for normal distribution first. For parametric data, 2-sample student *t*-test and one-way analysis of variance (ANOVA) plus Tukey's post-hoc comparison were used to compare differences among two or multiple groups, respectively. Multiple *t*-test was used to make comparisons in case of multiple comparison pairs co-exist, with Holm−Šídák method for correction. For nonparametric data, Kolmogorov−Smirnov test was used. A significant level was defined when $P < 0.05$, All statistical analysis and data figure plotting were performed by GraphPad Prism 7.0 (La Jolla, CA, USA).

## Reporting summary
Further information on research design is available in the Nature Portfolio Reporting Summary linked to this article.

## Data availability
All data generated or analyzed during this study are included in this published article (and its supplementary information files). Source data are provided with this paper.

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

## Acknowledgements

We thank Dr. Xuanjun Liu (The First Affiliated Hospital, Jinan University) in recruiting MDD patients for blood sample collection. This study was funded by STI2030-Major Projects (2022ZD0207600) and National Key Research and Development Program of China (2020YFA0113600) to L.Z., National Natural Science Foundation of China (32070955 to L.Z., and U22A20301 to K-F.S.), Guangdong Basic and Applied Basic Research Foundation (2023B1515040015) to L.Z., Guangdong Major Project of Basic and Applied Basic Research (2023B0303000004) to K.F.S. and L.Z., The Key Research and Development Plan of Ningxia (2022BEG01004) to K.F.S., Science and Technology Program of Guangzhou, China (SL2023A03J00544) to L.Z., and Science and Technology Program of Guangzhou, China (202007030012) to K-F.S. and L.Z.

## Author contributions

L.Y. and Y.W. performed 2-photon imaging, animal surgery, and stereotaxic injection, and analyzed experimental data. F.Y. executed molecular assays and M.W. worked in the immunostaining, with assistance from D.Y. and W.W. The patient recruiting and blood sample collection was performed by Y.J. The manuscript was written by L.Y. and L.Z., with inputs from all authors, and was revised by L.Z. and L.S. K-F.S. and L.Z. supervised this work. The funding was provided by K-F.S, L.S., and L.Z. All authors have read and approved this manuscript.

## Competing interests

All authors declare no competing interests.
