## [Peer Review File · Nature Communications]

Stress increases hepatic release of lipocalin-2 which contributes to anxiety-like behaviour in miceEditorial Note: This manuscript has been previously reviewed at another journal. This document only contains reviewer comments and rebuttal letters for versions considered at *Nature Communications*.

REVIEWERS' COMMENTS

Reviewer #1 (Remarks to the Author):

I have previously reviewed this manuscript and so, here, I will not reiterate my remarks on the summary of the work, the noteworthy results and its significance. Please see my previous comments in the Response to Referees Letter/Related Manuscript File.

The authors have since addressed all of my comments and I believe the manuscript is much improved with regards to ease of reading/understanding (reference to figures, etc.) and in terms of clarity in a number of areas.

They have removed what they referred to as the 'controversial' sections - which included the attempt to prove direct vagal efferent innervation of the liver which both reviewer 3 and I had issues with.

Subsequently, this has been correlated with the inclusion of a number of statements throughout the manuscript acknowledging there is not full evidence of this direct hepatic efferent innervation by vagus.

However, I would suggest that this is clarified in the abstract too as the way it is written now suggests to a reader that this novel brainstem-liver-cortical loop is fully described and evidence provided. Only later do you then see that it is not fully proven. Something in line with the statement added in lines 269 would do.

Otherwise, I understand this manuscript to contain a lot of interesting experiments and findings pertaining to the role of LCN2 in anxiety disorders and the cross-talk between the liver and brain under environmental stress.

I recommend for publication.

Reviewer #1 (Remarks to the Author):

I have previously reviewed this manuscript and so, here, I will not reiterate my remarks on the summary of the work, the noteworthy results and its significance. Please see my previous comments in the Response to Referees Letter/Related Manuscript File.

The authors have since addressed all of my comments and I believe the manuscript is much improved with regards to ease of reading/understanding (reference to figures, etc.) and in terms of clarity in a number of areas.

They have removed what they referred to as the 'controversial' sections - which included the attempt to prove direct vagal efferent innervation of the liver which both reviewer 3 and I had issues with.

Subsequently, this has been correlated with the inclusion of a number of statements throughout the manuscript acknowledging there is not full evidence of this direct hepatic efferent innervation by vagus.

However, I would suggest that this is clarified in the abstract too as the way it is written now suggests to a reader that this novel brainstem-liver-cortical loop is fully described and evidence provided. Only later do you then see that it is not fully proven. Something in line with the statement added in lines 269 would do.

Otherwise, I understand this manuscript to contain a lot of interesting experiments and findings pertaining to the role of LCN2 in anxiety disorders and the cross-talk between the liver and brain under environmental stress.

I recommend for publication.

A: Thank you for the efforts in reviewing our manuscript. For your advice regarding the content of the title and abstract, we have made changes accordingly, to remove the relevant description of the direct brain-liver loop. The new title is "Stress increases hepatic release of lipocalin-2 which contributes to anxiety-like behaviour in mice", and the abstract has also been modified.